# Weight Gain after Interferon-Free Treatment of Chronic Hepatitis C—Results from the German Hepatitis C-Registry (DHC-R)

**DOI:** 10.3390/biomedicines9101495

**Published:** 2021-10-19

**Authors:** Bernhard Schlevogt, Klaus H. W. Boeker, Stefan Mauss, Hartwig Klinker, Renate Heyne, Ralph Link, Karl-Georg Simon, Christoph Sarrazin, Yvonne Serfert, Michael P. Manns, Heiner Wedemeyer

**Affiliations:** 1Department of Medicine B, Münster University Hospital (UKM), 48149 Münster, Germany; 2Center of Hepatology, 30161 Hannover, Germany; prof.boeker@leberpraxis-hannover.de; 3Center for HIV and Hepatogastroenterology, 40237 Düsseldorf, Germany; stefan.mauss@center-duesseldorf.de; 4Department of Internal Medicine II, University Hospital Würzburg, 97080 Würzburg, Germany; Klinker_H@ukw.de; 5Leberzentrum am Checkpoint, 10961 Berlin, Germany; heyne@leberzentrum-checkpoint.de; 6MVZ-Offenburg GmbH/St. Josefs-Klinik, 77654 Offenburg, Germany; ralph.link@mvz-offenburg.de; 7MVZ Dres. Eisenbach, Simon, Schwarz GbR, 51375 Leverkusen, Germany; kg.simon@gastroenterologie-leverkusen.de; 8St. Josefs-Hospital, Medical Clinic 2, 65189 Wiesbaden, Germany; csarrazin@joho.de; 9Department of Internal Medicine 1, Goethe University Hospital Frankfurt, 60590 Frankfurt am Main, Germany; 10Leberstiftungs-GmbH Deutschland, 30625 Hannover, Germany; serfert.yvonne@mh-hannover.de (Y.S.); wedemeyer.heiner@mh-hannover.de (H.W.); 11Department of Gastroenterology, Hepatology and Endocrinology, Hannover Medical School, 30625 Hannover, Germany; Manns.Michael@mh-hannover.de

**Keywords:** chronic hepatitis C, direct-acting antivirals, interferon-free, HCV cure, weight gain, German Hepatitis C-Registry

## Abstract

Chronic hepatitis C can be treated very effectively with direct-acting antivirals (DAA) with only minor side effects compared to an interferon-containing treatment regimen. The significance of metabolic comorbidities after HCV cure is not well defined. This study aims to investigate short- and long-term weight change of patients receiving interferon-free antiviral treatment for chronic hepatitis C. The German Hepatitis C-registry (DHC-R) is a national multicenter real-world cohort. A total of 5111 patients were followed prospectively after DAA treatment for up to 3 years. Weight change compared to baseline was analyzed at end of treatment and at years 1, 2, and 3 after completion of antiviral therapy. Regression analysis was performed to identify baseline predictors for weight change. While there was no relevant mean weight change (−0.2 kg, SD 4.3 kg) at the end of antiviral treatment, weight started to increase during long-term follow-up reaching +1.7 kg (SD 8.0 kg, *p* < 0.001) compared to baseline at 3 years (follow-up year 3, FU3) after completion of antiviral therapy. 48%, 31%, and 22% of patients had a weight gain greater than 1, 3, and 5 kg at FU3, respectively. During follow-up, a body mass index (BMI) <30 proved to be the only consistent predictor for weight gain. DAA treatment is followed by a substantial weight gain (+3 kg or more) in one-third of the patients during long-term follow-up. Non-obese patients seemed to be most vulnerable to weight gain. The body compartment involved in weight gain as well as the mechanism of weight gain remain to be elucidated.

## 1. Introduction

The era of direct-acting antivirals (DAA) has revolutionized treatment of chronic hepatitis C virus (HCV) infection. Excellent treatment effectiveness, safety, and a risk reduction in liver cirrhosis-related complications can be achieved [1]. Apart from this, a great beneficial impact on the spectrum of extrahepatic manifestations was observed after interferon-free HCV cure [2]. Patient-reported outcomes have improved after sustained virological response following DAA treatment [3]. Furthermore, various changes in lipid and glucose metabolism can be observed after interferon-free HCV cure [4], such as an improved glycemic control in diabetic patients [5].

During the interferon era, those aspects of HCV cure that could not be studied as interferon therapy itself was a confounder due to its severe side effects. Interferon-based HCV treatment was associated with weight loss in up to 90% of patients, although most patients regained their initial weight after completion of therapy [6,7]. Treatment with DAA is regarded to be virtually free of relevant side effects so the true effect of HCV cure can now be studied.

Small single center studies and one large United States veteran cohort study suggest substantial weight gain following successful DAA treatment [8,9,10]. In line with these results, liver steatosis, measured noninvasively by controlled attenuation parameter (CAP) was shown to increase significantly after interferon-free HCV cure [11,12]. As a potential concern, liver steatosis before and after HCV cure was associated with impaired regression or even progression of liver fibrosis [13,14].

The aim of this study was to determine the short-term and long-term effects of interferon-free cure of chronic hepatitis C on body weight on a large scale, analyzing data from the German Hepatitis C-Registry which is a prospective nationwide real-world cohort.

## 2. Methods

The DHC-R (German Hepatitis C-Registry) is a national multicenter non-interventional real-world registry currently including about 18,000 patients recruited by more than 250 centres (BfArM-ID: 2493; DRKS-ID:DRKS00009717). The study is conducted in accordance with the Declaration of Helsinki and Good Clinical Practice guidelines and was approved by the Institutional Review Board (Ethics Committee of Ärztekammer Westfalen-Lippe; reference number 2014-395-f-S). Patients had to provide written informed consent. Adult patients (≥18 years of age) of both genders with chronic hepatitis C virus (HCV) infection and detectable HCV RNA were included. The main exclusion criteria were pregnancy (patient or female partner of male patient), women in nursing period or women of childbearing age without reliable contraception as well as contraindications for use of antiviral treatment. The choice of antiviral therapy was at the discretion of the physician. Patient data were collected by a web browser-based Electronic Data Capture (EDC) system without software installation on site (BEO, e.factum GmbH, Butzbach, Germany) hosted at a Clinical Research Organization (CRO). Data were checked for plausibility as well as on site monitoring.

For the present analysis, patients receiving a combination therapy of DAA with pegylated interferon were excluded. Patient reported weight had to be available at DAA treatment initiation and at least one further time point (end of treatment (EOT), follow-up week 24 (FU24), year 1 (FU1), year 2 (FU2) and year 3 (FU3)). Weight change was defined as compared to baseline weight.

### Statistical Analysis

Data were analyzed as of 15 July 15 2018. Summary statistics, frequencies, and proportions were assessed dependent on the scale level of the data. *p*-values for weight change were analyzed by paired samples test (2-tailed). Factors for weight change were analyzed by univariate and multivariate analysis (Anova 2-sided). Significant univariate parameters were included in the multivariate analysis if data were available for at least 75% of the patients. Analyses were calculated using SPSS (IBM Corp. Released 2020. IBM SPSS Statistics for Windows, Version 22.0. Armonk, NY: IBM Corp).

## 3. Results

Weight change compared to baseline weight could be calculated at EOT for 5111 patients, at FU24 for 4442 patients, at FU1 for 1398 patients, at FU2 for 1405 patients, and at FU3 for 496 patients, respectively. Patient’s characteristics of all mentioned cohorts are shown in Table 1. Around 40% of patients were female, 25% were cirrhotic, and 5.4% had a hepatic decompensation. Mean age for the EOT cohort was 52.4 years (±12.7 years). At baseline, 18% of patients were obese (BMI > 30), whereas only 2.4% where underweight (BMI < 18.5). Ribavirin was added to DAA treatment in 32% of the cases. Due to prioritization of treatment for patients with advanced liver disease shortly after DAA approval, all different follow-up cohorts showed slightly different baseline characteristics. Sustained virological response at FU24 could be achieved in 96% of patients.

At EOT, mean weight change compared to baseline was −0.2 kg, standard deviation (SD) was 4.3 kg. After completion of DAA treatment, the mean weight started to gradually increase (FU24: 0.3 kg, SD 5.2 kg, FU1: 0.7 kg, SD 6.3 kg, FU2: 1.6 kg, SD 6.6 kg, FU3: 1.7 kg, SD 8.0 kg), reaching a maximum weight gain at FU3 (Figure 1). All weight changes were statistically significant compared to baseline with a *p*-value below 0.001.

To further illustrate weight development during and after DAA treatment, the proportion of patients with weight gain, weight loss, and stable weight were analyzed for the different follow-up intervals. Weight change was defined as a change of more than 1 kg, 3 kg, and 5 kg compared to baseline. In Figure 2A, weight gain was defined as a change of more than 1 kg. At EOT, 21.1% of patients had weight gain. This number increased markedly during follow-up to 47.8% with weight gain 3 years after completion of DAA therapy. However, when looking at weight loss, this proportion remained stable at around 25% throughout all follow-up intervals (Figure 2A). As a weight change of just more than 1 kg might be considered irrelevant, we calculated the proportion of patients with a weight change greater than 3 and 5 kg (Figure 2B,C). Obviously, these definitions reduced the proportion of patients with weight change but the proportion of patients with weight gain increased during the follow-up intervals in a very similar way. At EOT, 9% of patients had a weight gain greater than 3 kg which increased to 31% at FU3. When looking at a weight gain greater than 5 kg, 4.1% were affected at EOT whereas 22% were affected at FU3. In Figure 2B,C the proportion of patients with weight loss showed a minor increase during follow-up but not comparable with the tremendous proportion of weight gain.

As weight gain occurred only in a subpopulation of DAA-treated patients we performed a regression analysis to identify predictors for weight gain in order to better characterize the weight gain cohort. Age ≥ 60 years, obesity, ribavirin treatment, sustained virological response (SVR), diabetes, cirrhosis, gender, Child-Pugh score, MELD score ≥ 15 and prior hepatic decompensation were included as independent variables. As weight gain was greatest at FU1, FU2, and FU3, univariate regression analysis was calculated at those time points (Table 2). Weight change in kilogram compared to baseline was the dependent variable. At FU1 and FU3 obesity, a BMI > 30 was the only variable which could predict weight gain, so no multivariate regression analysis had to be performed (FU1: BMI ≤ 30 versus > 30, coefficient −1.9, 95% confidence interval (CI) −2.7–−1.0, *p* < 0.0001; FU3: BMI ≤ 30 versus > 30, coefficient −3.4, 95% CI −5.2–−1.6, *p* < 0.0001). At FU2, there were two variables found to be significant predictors in univariate regression analysis, so multivariate regression analysis was performed, which confirmed obesity and gender as predictors (BMI ≤ 30 versus > 30, coefficient −1.7, 95% CI −2.6–−0.8, *p* < 0.0001; male versus female, coefficient −0.8, 95% CI −1.5–−0.06; *p* = 0.034). Obesity was inversely correlated with weight gain throughout FU1, FU2, and FU3 after DAA treatment for chronic HCV infection. As gender could only predict weight gain at FU2 (*p* = 0.034) and not at FU1 and FU3, gender does not seem to be a consistent predictor. The stage of liver disease, age, RBV treatment, SVR, and diabetes could not predict weight change in a statistically significant way.

To better demonstrate the relationship between obesity and weight gain during DAA treatment we stratified mean weight change compared to baseline to BMI ≤ 30 and BMI > 30 for all follow-up intervals (Figure 3). In patients with a BMI ≤ 30, mean weight gain compared to baseline was much greater than in the overall cohort and in the BMI > 30 cohort (BMI ≤ 30 FU3: 2.4 kg, SD 7.7 kg; BMI > 30 FU3: −1,0 kg, SD 8.9 kg; overall cohort FU3: 1.7 kg, SD 8.0 kg (Figure 1 and Figure 3). In contrast, patients with a BMI > 30 had a stable weight (FU1, FU2, and FU3) or even a minor weight loss compared to baseline (EOT, FU24). When comparing mean weight change between BMI ≤ 30 and BMI > 30, differences were statistically significant at all five follow-up intervals. Furthermore, weight gain was statistically significant compared to baseline in patients with a BMI ≤ 30 at FU24, FU1, F2, and FU3. There was no statistic significant weight change compared to baseline at FU1, FU2, and FU3 in the BMI > 30 cohort. Therefore, weight gain in the course of DAA treatment seems to occur preferentially in the subgroup of non-obese patients.

Appendix A shows mean weight change compared to baseline stratified by further BMI subdivision (underweight BMI < 18.5, healthy weight BMI 18.5–25, overweight BMI > 25–30, obese BMI > 30). Weight gain is clearly restricted to the healthy weight and overweight cohort, whereas underweight patients as well as obese patients did not show a relevant weight gain throughout follow-up. As a limitation, patient numbers in the underweight cohort were quite low (approximately 2.4% of all cohorts), making definite conclusions for this cohort unreliable. When comparing a combined cohort of healthy weight and overweight patients with underweight patients, weight change only reached significance level at FU2 whereas significance level was missed at FU1. FU3 was not examined due to insufficiently small sample size. However, the observed weight gain can obviously not be attributed to an extensive weight increase of underweight patients cured from chronic hepatitis C.

## 4. Discussion

In this large prospective national multicenter real-world registry study of DAA-treated patients with chronic hepatitis C, we show that a large proportion of patients substantially gained weight in the long-term follow-up. This weight gain started after completion of DAA therapy and increased during the follow-up for three years. Univariate and multivariate regression analysis revealed a BMI below 30 as the only consistent risk factor for weight gain, whereas patients with a BMI over 30 did not show a substantial weight gain. Of note, other parameters such as stage of liver disease did not predict weight gain. Weight gain was also not a phenomenon observed primarily in underweight patients.

Our study could confirm the observation of two single center studies and one large veteran cohort study describing weight gain in the course of DAA therapy [8,9,10]. As our study merely had a descriptive character, the mechanism of weight gain as well as the body compartment responsible for weight gain remains a matter of speculation.

In contrast to our study, Do et al. could identify more predictors of weight gain, such as SVR, cirrhosis, and obesity at baseline [10]. This contrasts with our findings, which might be explained by different characteristics of the patient cohort (e.g., predominant male patients, more obese patients at baseline). Still, further studies are needed to address this question.

Other groups have compared skeletal muscle mass by bioimpedance analysis and computer tomography before and after DAA treatment. They found an increase in skeletal muscle mass after DAA treatment [15,16] which correlated in one study with weight gain and a decrease of visceral fat. The increase in skeletal muscle mass occurred predominantly in patients with low muscle mass at baseline [17]. Importantly, sarcopenia in chronic hepatitis C seems to occur independently from the stage of liver disease [18].

Amelioration of inflammatory burden [19] as well as improved quality of life and reduced fatigue [3] have been reported to be a result auf DAA-induced HCV cure. This could therefore lead to increased physical activity and reversal of preexisting sarcopenia.

Weight gain as a result of an increased muscle mass could elegantly explain why obese patients in our study had no relevant weight gain as they might not have been able to sufficiently increase their physical activity after the HCV cure compared to non-obese patients.

All these mentioned studies had a short follow-up of no more than one year after DAA treatment. So, the body compartment involved in weight gain occurring strongest at FU2 and FU3 remains unknown.

Besides an increase in muscle mass, an increase in body fat as a cause of weight gain has to be considered. In line with this theory, several studies report an increase in liver steatosis, measured noninvasively after successful DAA therapy [11,12,13]. As a potential concern, liver steatosis might mitigate fibrosis reversal after HCV cure [13,14] and increase the risk for hepatocellular carcinoma [20]. Furthermore, obesity after HCV cure is likely to increase extrahepatic morbidity. In the DHC-R it could be shown that elevated liver enzymes after HCV cure were associated with steatosis. Importantly, steatosis associated enzyme elevation after HCV cure was associated with mortality [21].

Future studies with body composition analysis and long-term follow-up need to determine whether weight gain after DAA therapy for chronic hepatitis C is beneficial or not. To understand the mechanism of weight gain, a correlation with patient reported outcomes would be of great interest.

In HIV infection, a similar phenomenon has been known for a long time. Weight gain often occurs after initiation of highly active antiretroviral therapy. Weight gain correlates with the effectiveness of antiviral therapy and has been explained as a “return to health” with reversal of the catabolic effects of HIV infection [22]. In this condition, body fat seems to be the main compartment involved in weight gain, leading to increased risk for diabetes and cardiovascular disease [22].

Constant weight gain during aging is known to occur in the general population. At first sight, the extent of weight gain observed in our cohort is not far away from this natural phenomenon, as in several German cohorts, annual weight gain in the general population is between 299–429 g [23]. To the best of our knowledge, natural weight change throughout aging in patients with chronic hepatitis C has not been studied. It is likely, though, that weight gain will occur to a lesser degree here than in the general population. Considering this and the fact that mean weight gain at FU3 in non-obese patients (BMI < 30) was 2.4 kg we do not believe we observed a natural weight gain in our study. Furthermore, in our study initially there was no mean weight change at end of treatment and afterwards, a constantly increasing proportion of patients with weight gain was observed. These dynamics of weight change argue against a natural occurring weight gain.

Our study has several limitations. Ideally, weight change should be studied in comparison to a matched cohort of untreated patient of chronic hepatitis C. However, to our knowledge there is no historic cohort of such patients with a similar follow-up period. Withholding HCV treatment for several years to create an untreated control group now seems to be unethical as basically all patients can be successfully treated and deterioration of liver function is a constant threat.

A further limitation is the fact that patient numbers decrease throughout the follow-up. As patients were included successively in this real-world registry, a long follow-up is not available for all patients. This study design was chosen to keep the cohort size high in order to allow for precise description of weight gain following DAA therapy. However, as we have shown in Table 1, baseline characteristics for each cohort did not differ very much, which makes a relevant bias unlikely. Furthermore, regression analysis should correct for minor differences in cohort composition.

At the first glance, it might seem illogical that SVR was not a significant predictor for weight gain in our study. However, patient numbers with non-SVR were very low (FU1: *n* = 60; FU2: *n* = 44; FU3: *n* = 9), making meaningful conclusions extremely doubtful. This is not surprising as most non-SVR patients will not have been observed for years but retreated with a more potent DAA regimen. Besides, even temporary viral suppression might have an effect on body weight.

Our study benefits from several strengths. Unselected large real-world cohort studies such as our study are usually far more representative than randomized controlled trials with highly selected patients. Following weight change for up to 3 years after DAA treatment is the longest follow-up period in this setting reported so far. This is particularly important as weight gain is largest at follow-up year 3. Unlike the study by Do et al. [10] using just one follow-up interval, we present four different follow-up intervals (EOT, FU1, FU2, FU3) which allows to better define the dynamics of weight change. Especially the direct effect of DAA therapy can be assessed by reporting end of treatment weight.

## 5. Conclusions

In conclusion, we reported a substantial body weight gain in the follow-up after interferon-free antiviral treatment for chronic hepatitis C. This weight gain predominantly occurred in patients with a body mass index below 30. Further studies are urgently needed to examine which body compartment is involved in weight gain. Depending on this issue, weight gain could be beneficial or not.

## Collaborators

Rainer Ullrich, Tim Zimmermann, Tobias Goeser, Willibold Schiffelholz, Margareta Frank-Doss, Christoph Antoni, Andreas Schober, Thomas Lutz, Maria-Christina Jung, Hjördis Möller, Anita Pathil, Michael R. Kraus, Heribert Knechten, Dietrich Hüppe, Markus Cornberg, Uwe Naumann, Martin Hoffstadt, Wolfgang Syska, Jan-Ulrich Sonne, Manfred Nowak, Stefan Scholten, Nazifa Qurishi, Katharina Willuweit, Katrin Ende, Gerd Klausen, Ingolf Schiefke, Iris Peuser, Birgit Kallinowski, Stefan Zeuzem, Thomas Berg, Peter Buggisch, Heinz Hartmann, Claus Niederau, Ulrike Protzer, Peter Schirmacher

## Figures and Tables

**Figure 1 biomedicines-09-01495-f001:**
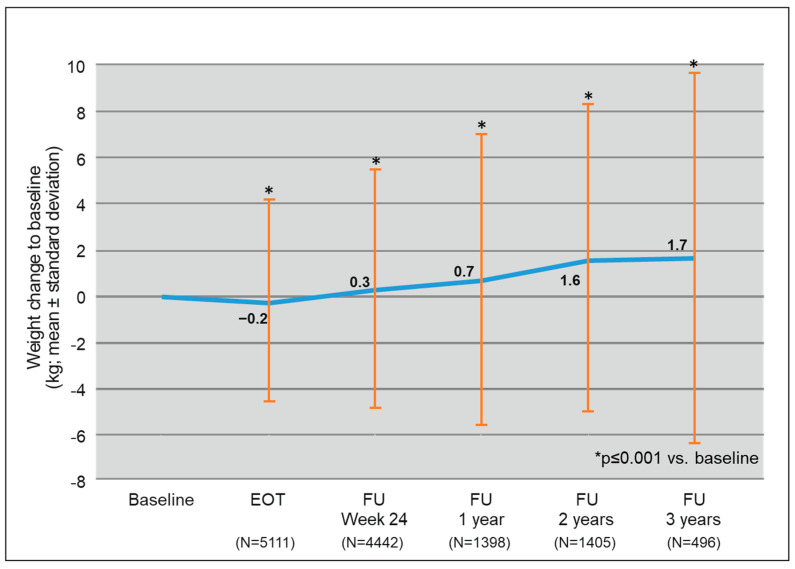
Increase in mean weight compared to baseline during follow-up of DAA Treatment. Abbreviations: EOT, end of treatment; FU24, follow up 12 to 24 weeks after EOT; FU1, follow up one year after EOT; FU2, follow up two years after EOT; FU3, follow up three years after EOT.

**Figure 2 biomedicines-09-01495-f002:**
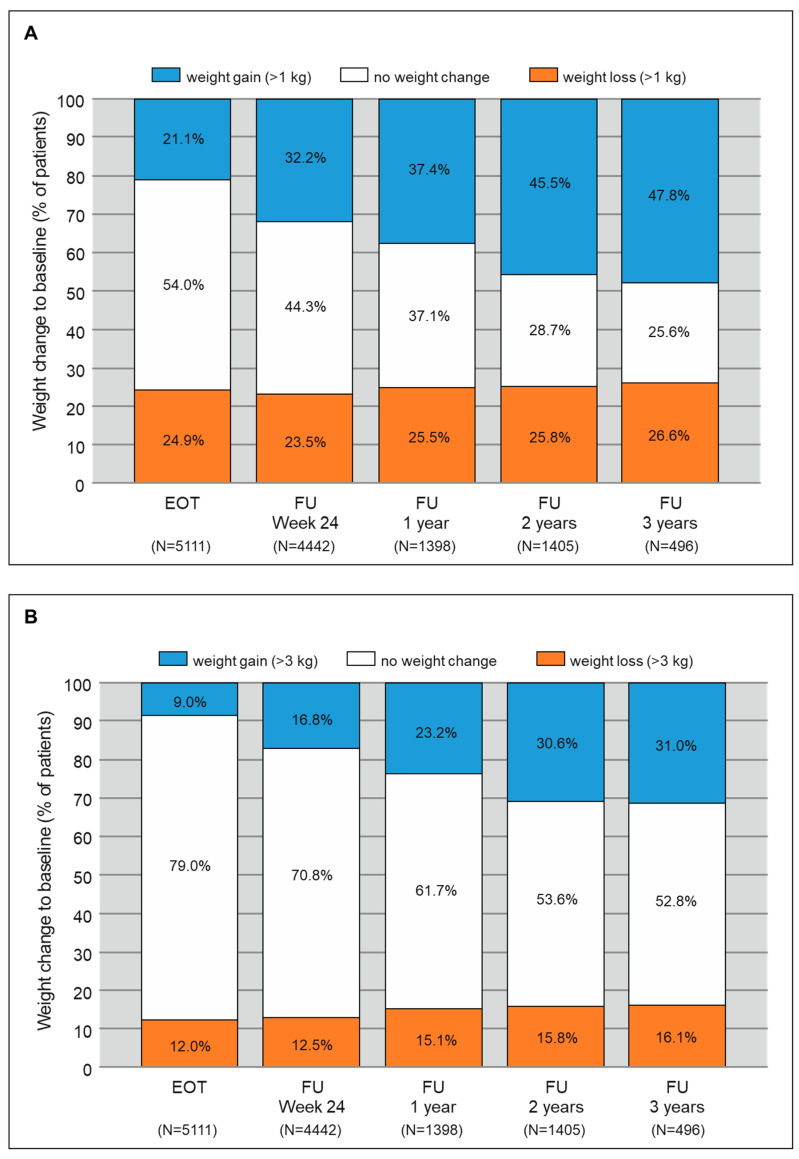
Proportion of patients with weight gain > 1 kg (**A**), > 3 kg (**B**), and > 5 kg (**C**) compared to baseline after DAA treatment. Abbreviations: EOT, end of treatment; FU24, follow up 12 to 24 weeks after EOT; FU1, follow up one year after EOT; FU2, follow up two years after EOT; FU3, follow up three years after EOT.

**Figure 3 biomedicines-09-01495-f003:**
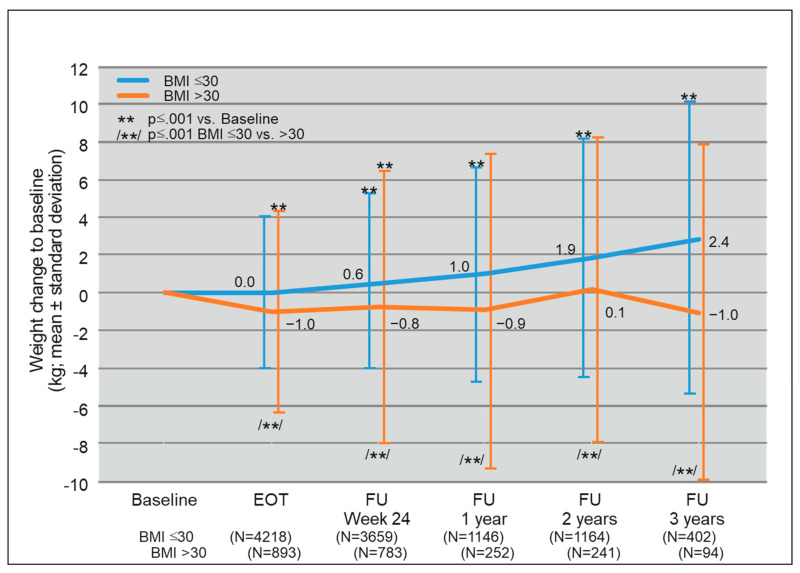
Mean weight change compared to baseline after DAA treatment stratified by obese and non-obese patients. Abbreviations: BMI, body mass index; EOT, end of treatment; FU1, follow up one year after EOT; FU2, follow up two years after EOT; FU3, follow up three years after EOT.

**Table 1 biomedicines-09-01495-t001:** Baseline characteristics and SVR rates of the EOT, FU24, FU1, FU2, FU3 cohorts of DAA treated patients in the German Hepatitis C-Registry.

Baseline Characteristic/SVR rate	EOT Cohort N = 5111	FU24 Cohort N = 4442	FU1 Cohort N = 1398	FU2 Cohort N = 1405	FU3 Cohort N = 496
Female sex, % (*n*)	39.5 (2021)	40.4 (1794)	38.4 (537)	41.1 (577)	40.9 (203)
Mean age, years ± SD	52.4 ± 12.7	53.1 ± 12.6	53.9 ± 12.5	54.5 ± 12.2	54.6 ± 11.7
Age < 60 years, % (*n*)	71.6 (3662)	69.8 (3102)	68.3 (955)	66.5 (934)	67.5 (335)
Liver cirrhosis, % (*n*)	25.2 (1289)	26.5 (1178)	34.9 (488)	34.0 (478)	40.9 (203)
Child Pugh Score A, % (*n*/N)	70.0 (902/1289)	69.5 (819/1178)	71.7 (350/448)	72.4 (346/478)	72.4 (147/203)
Hepatic decompensation, % (*n*)	5.4 (275)	5.1 (225)	8.2 (114)	7.8 (109)	8.5 (42)
MELD Score > 15, % (*n*)	4.0 (50)	3.5 (37)	5.5 (21)	3.8 (17)	3.2 (6)
BMI < 18.5 kg/m^2^, % (*n*)	2.3 (118)	2.2 (97)	2.6 (37)	2.4 (34)	2.4 (12)
BMI 18.5–15 kg/m^2^, % (*n*)	45.3 (2313)	44.8 (1989)	46.1 (645)	45.2 (6235)	40.9 (203)
BMI > 25–30 kg/m^2^, % (*n*)	35.2 (1799)	35.7 (1586)	33.5 (468)	35.5 (499)	37.9 (188)
BMI > 30 kg/m^2^, % (*n*)	17.2 (881)	17.3 (770)	17.7 (248)	16.9 (237)	18.8 (93)
Diabetes mellitus, % (*n*)	9.1 (465)	10.4 (461)	13.6 (190)	12.9 (181)	15.9 (79)
SVR, % (*n*)	*n*/a	96.0 (4243/4418)	95.7 (1332/1392)	96.6 (1355/1399)	98.2 (485/494)

Abbreviations: BMI, body mass index; EOT, end of treatment; FU24, follow up 12 to 24 weeks after EOT; FU1, follow up one year after EOT; FU2, follow up two years after EOT; FU3, follow up three years after EOT; HCV, hepatitis C virus; MELD, Model of End Stage Liver Disease; *n*/a, not applicable, SD, standard deviation; SVR, sustained virological response.

**Table 2 biomedicines-09-01495-t002:** Univariate und multivariate regression analysis for predictors of weight gain in the course of DAA treatment at FU1, FU2, and FU3.

Follow-Up Year 1—Univariate Analysis	Coefficient	*p*-Value	95% CI for Coefficient
no SVR vs. SVR	1.422	0.086	−0.200–3.044
non-cirrhotic vs. cirrhotic	0.240	0.496	−0.450–0.930
no diabetes vs. diabetes	−0.019	0.969	−0.979–0.940
male vs. female	−0.460	0.182	−1.136–0.216
no RBV vs. RBV	−0.069	0.842	−0.752–0.613
no hepatic decomp. vs. hepatic decomp.	0.275	0.653	−0.926–1.477
<60 years vs. ≥60 years	−0.214	0.554	−0.920–0.493
BMI ≤ 30 vs. > 30	−1.859	0.000	−2.709–−1.009
CHILD A vs. other	0.002	0.998	−1.592–1.596
MELD < 15 vs. ≥ 15	−1.582	0.293	−4.537–1.373
**Follow-Up Year 2—Univariate Analysis**	**Coefficient**	** *p* ** **-Value**	**95% CI for Coefficient**
no SVR vs. SVR	1.029	0.309	−0.953–3.012
non-cirrhotic vs. cirrhotic	0.300	0.419	−0.429–1.030
no diabetes vs. diabetes	−0.328	0.533	−1.360–0.704
male vs. female	−0.875	0.014	−1.577–−0.174
no RBV vs. RBV	0.166	0.648	−0.548–0.880
no hepatic decomp. vs. hepatic decomp.	0.982	0.136	−0.310–2.274
<60 years vs. ≥60 years	−0.637	0.088	−1.368–0.095
BMI ≤ 30 vs. > 30	−1.811	0.000	−2.724–−0.899
CHILD A vs. other	−1.581	0.135	−3.657–0.495
MELD < 15 vs. ≥ 15	−0.716	0.691	v4.254–2.822
**Follow-Up Year 2—Multivariate Analysis**			
male vs. female	−0.759	0.034	−1.460–−0.058
BMI ≤ 30 vs. > 30	−1.724	0.000	−2.639–−0.809
**Follow-Up Year 3—Univariate Analysis**	**Coefficient**	** *p* ** **-Value**	**95% CI for Coefficient**
no SVR vs. SVR	−1.273	0.637	v6.578–4.031
non-cirrhotic vs. cirrhotic	0.054	0.942	−1.384–1.492
no diabetes vs. diabetes	−1.160	0.238	−3.090–0.769
male vs. female	−1.150	0.116	−2.585–0.284
no RBV vs. RBV	−0.412	0.571	−1.843–1.018
no hepatic decomp. vs. hepatic decomp.	1.334	0.302	−1.203–3.871
<60 years vs. ≥60 years	−0.312	0.685	−1.822–1.197
BMI ≤ 30 vs. > 30	−3.402	0.000	−5.181–−1.624
CHILD A vs. other	1.213	0.552	−2.810–5.236
MELD < 15 vs. ≥ 15	1.810	0.616	−5.300–8.920

Abbreviations: BMI, body mass index; CI, confidence interval; EOT, end of treatment; FU1, follow up one year after EOT; FU2, follow up two years after EOT; FU3, follow up three years after EOT; MELD, Model of End Stage Liver Disease; RBV, ribavirin; SVR, sustained virological response.

## Data Availability

The data presented in this study are available on request from the corresponding author. The data are not publicly available due to privacy reasons.

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
