# Peer review of "Weight Gain after Interferon-Free Treatment of Chronic Hepatitis C—Results from the German Hepatitis C-Registry (DHC-R)"

_biomedicines, 2021, doi:10.3390/biomedicines9101495_

Round 1
Reviewer 1 Report
Comments:
- The decreasing numbers of study patients over the follow-up time (from 5,111 to 469, i.e. a loss of about 90 % of the patients) may be the basis for relevant bias of the results. This weakness of the study is not mentioned and not discussed by the authors.
- With relevant weight gain only in the group of patients with "BMI≤30" - which is a very heterogeneous group- the question arises whether weight gain may only have occurred in primarily underweight patients. Therefore, the authors should subdive this group further, e.g. <21, 21-25 and 25-30 - or as they consider appropriate. This would definitely help with interpretation of their findings.
Reviewer 2 Report
The study is well done, the material is large enough and the methods look reliable. However the study is based on extensive and very recent literature, gives some new information and this warrants its publication.
Round 2
Reviewer 1 Report
adequate response from authors